# Validation of self-reported hypertension in young adults in the US-based Growing Up Today Study (GUTS)

Jie Chen[1,2]*, Jaime E. Hart[1,2], Naomi D. L. Fisher[3], Francine Laden[1,2,4]

**1** Department of Environmental Health, Harvard T. H. Chan School of Public Health, Boston, Massachusetts, United States of America, **2** Channing Division of Network Medicine, Department of Medicine, Brigham and Women's Hospital and Harvard Medical School, Boston, Massachusetts, United States of America, **3** Division of Endocrinology, Diabetes and Hypertension, Brigham and Women's Hospital, Boston, Massachusetts, United States of America, **4** Department of Epidemiology, Harvard T. H. Chan School of Public Health, Boston, Massachusetts, United States of America

* jiechen@hsph.harvard.edu

## Abstract

### Background

Self-reporting is often used in large epidemiologic research to identify hypertensive participants, but its validity in young adults has not been sufficiently assessed. We aimed to validate self-reported hypertension diagnosis from questionnaires by using medical records among a sample of young adults aged 22–39 years in the Growing Up Today Study (GUTS), during a time of transition in national hypertension definitions.

### Methods

A sample of 1,000 GUTS participants were asked for permission to access their medical records, to confirm their self-reported hypertension status on questionnaires from 2010 to 2019. Paired self-reported and medical record information was available for 318 participants. Medical records were reviewed by a clinical hypertension specialist. We evaluated agreement, kappa statistic, sensitivity and specificity of self-reporting. We assessed the correlations between blood pressure measurements self-reported on the 2019 questionnaire and those from medical records and compared the measurements using paired t-tests.

### Results

The selected sample was generally representative of the full GUTS cohort. Agreement, kappa coefficient, sensitivity and specificity of self-reported hypertension were 85.5%, 0.72, 100%, and 75.3%. Although the absolute differences in blood pressure measurements between self-report and medical records were small (e.g., the average difference in typical recent blood pressures was 3.5/1.2 mm Hg), these measures were only moderately correlated.

**Data Availability Statement:** Data cannot be shared publicly because of regulations that protect participants' personal information. Researchers interested in obtaining access to the Growing Up Today Study (GUTS) data will be required to

submit an external investigator form (https://gutsweb.org/collaborate-with-guts/). Analysis codes are available upon request from the corresponding author or the Study committee (guts@channing.harvard.edu).

**Funding:** This study was supported by NIH grants U01 HL145386, U01 CA176726, R01 ES029840, and P30 ES000002. The funders had no role in study design, data collection and analysis, decision to publish, or preparation of the manuscript.

**Competing interests:** The authors have declared that no competing interests exist.

## Conclusions

Validity of self-reported hypertension was high in GUTS, ensuring use as an endpoint in future studies with confidence. We demonstrated that young adults likely without formal medical training are able to report hypertension status with reasonable accuracy.

## Introduction

Self-reporting is a simple and low-cost method that is often used in large epidemiologic studies for identifying participants with hypertension [1, 2]. The validity of self-reported hypertension can be influenced by demographic, socioeconomic, and cultural factors [3–6]. A meta-analysis of 11 studies found that self-report correctly identified individuals with hypertension in 42% (95% confidence interval [CI], 31–54) of hypertensive cases and correctly identified individuals without hypertension in 90% (95% CI 84–93) of non-cases [4]. Large heterogeneity was found across individual studies. Using the India National Family Health Survey data, researchers found self-reported hypertension has higher sensitivity among individuals aged 35–49 years than those aged 15–34 years, while specificity was higher among the younger individuals [6]. The younger adults may differ from the older populations in many aspects that could affect the validity of self-reporting, such as frequency of doctor visits and awareness of health status. While the majority of the previous research has been conducted in older populations [5, 7], the validity of self-reported hypertension in young adults has not been sufficiently assessed.

The Growing Up Today Study (GUTS) was established to study the health of adolescents and young adults [8]. GUTS I and GUTS II (known collectively as GUTS) were founded in 1996 and 2004, respectively, by inviting mothers in the ongoing Nurses' Health Study II (NHS II) to enroll their children who were then between the ages of 9 and 14 (GUTS I) or 10 and 17 (GUTS II) [9]. More than 27,000 participants across the US completed and returned the baseline questionnaires. Follow-up questionnaires were sent to participants annually. The cohorts were combined in 2013 when all participants had reached age 18. Approximately 86% of the females and 83% of the males have returned at least one follow-up questionnaire. Self-reported hypertension has been asked on all questionnaires since 2010. To use self-reported hypertension data with confidence, it is necessary to examine its validity.

Therefore, we aimed to validate hypertension diagnoses and blood pressure measurements from self-reported questionnaires using medical records among a sample of participants in GUTS. In the current study, we included follow-up data up to the end of 2019, covering the period of transition in the US national definitions of hypertension from 140/90 millimeters of mercury (mm Hg) to 130/80 mm Hg [10, 11].

## Materials and methods

### Study design

We identified a random sample of 500 participants who ever reported being diagnosed with hypertension, and a random sample of 500 participants who never reported a hypertension diagnosis, based on questionnaires from 2010 to 2019. We asked the 1000 participants for permission to access their medical records to confirm the self-reported information. Of the 1000 participants invited, 201 (20.1%) provided written consent to the study and their medical records were scanned for review. All participants were recruited prospectively from June 23rd 2020 to March 9th 2022. An additional 117 GUTS participants previously provided written

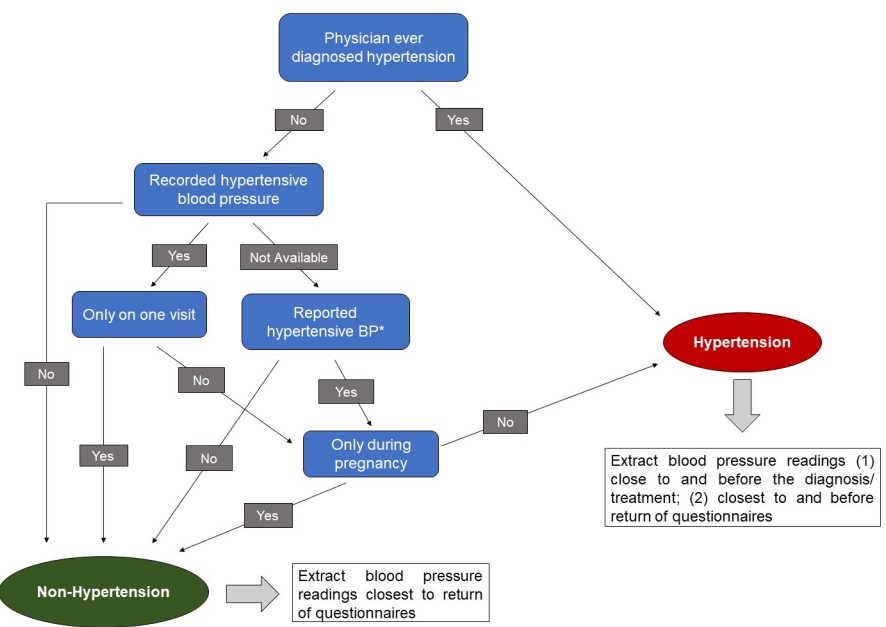

**Fig 1. Diagnosis of hypertension through physician review of medical records.** * Acknowledgment of hypertensive blood pressure without blood pressure readings recorded.

consent to provide their medical records for hypertension status confirmation, and thus paired self-reported and medical record information was available for 318 participants in the current analyses. This study was approved by the Institutional Review Board at the Brigham and Women's Hospital and participants provided informed written consent.

Hypertension was self-reported on all questionnaires from 2010 to 2019. The question asks, "Have you ever been diagnosed with high blood pressure or hypertension?" which was adapted from a question asked in Wave III of the National Longitudinal Study of Adolescent to Adult Health (Add Health) [12]. Using this question, hypertension prevalence was estimated to be 5.6% in 15,187 young adult (18–26 years) participants in Add Health, which was close to estimates from studies where classification of hypertension was based on measured blood pressure [13]. As a follow-up question, GUTS participants were asked to provide the year of first diagnosis. In the 2019 questionnaire, participants were further asked to recall their most recently measured (within 2 years) systolic and diastolic blood pressure. Participants who were on blood pressure lowering medications were additionally asked to recall their office blood pressure prior to starting medication.

We requested medical records from the clinician reported by the participant to have provided their hypertension diagnosis, starting from the reported date of diagnosis to the most recent record available. For participants who did not report a hypertension diagnosis, we requested medical records from their primary care physician closest to return of the questionnaire. Through physician (NF) review of the medical records, we defined participants as having the diagnosis of hypertension when a physician recorded the diagnosis of hypertension in a medical note or problem list or recorded hypertensive blood pressure (SBP/DBP $\geq$140/90 mm Hg prior to 2018 or $\geq$130/80 mm Hg afterwards) more than once (Fig 1). We defined participants as not having the diagnosis of hypertension if there was no diagnosis noted and no indication of any treatment of hypertension, if a problem list was present and did not include hypertension, or if a physician indicated hypertension based on high blood pressure on only

one visit and blood pressure was verified normal on future visits. Additionally, because our focus was on the analysis of chronic hypertension, if participants had hypertension diagnosed only during pregnancy and not prior to pregnancy or post-delivery, they were not defined as hypertensive. Date of hypertension diagnosis was taken from the medical record if available (95%), or otherwise from patient report. When available, we extracted blood pressure readings for hypertensive patients from the medical record at two time points: 1) closest to and before the diagnosis or before treatment initiation, and 2) closest to and before return of the questionnaire, up to five years previous. For non-hypertensive participants, we extracted blood pressure readings closest to return of the questionnaire, up to five years previous.

## Statistical analysis

We calculated agreement, sensitivity, and specificity of the self-reported hypertension measure. Sensitivity reflects the ability of self-reporting to identify participants with hypertension correctly; specificity reflects the ability of self-reporting to identify participants without hypertension correctly. We also calculated Cohen's kappa coefficient, which is an adjusted agreement measure taking into account the possibility of chance agreement [14]. As suggested by Cohen, we interpreted the kappa result as follows: values $\leq 0$ as indicating no agreement, 0.01–0.20 as none to slight, 0.21–0.40 as fair, 0.41–0.60 as moderate, 0.61–0.80 as substantial, and 0.81–1.00 as almost perfect agreement. We evaluated the same metrics in subgroups of males and females to determine if there were any sex-related differences in reporting. Statistical significance testing was performed using Chi-squared tests or Fisher's exact tests as appropriate.

We assessed the correlations between blood pressure readings self-reported on the 2019 questionnaire and those recorded in the medical records using both Pearson and Spearman's correlation coefficients. We compared blood pressure readings from two sources using paired t-tests. We reported the mean difference and 95% confidence intervals (CIs) based on an $\alpha = 0.05$. A 95% CI not including zero indicates a statistically significant difference between blood pressure readings. We further created a Bland-Altman plot to assess the agreement between the two measurements. For any one visit, only the first reading from the medical record was used in the analyses because blood pressure was most often measured only once; only 20 out of 230 people had blood pressure readings recorded twice at one visit.

## Results

Table 1 shows characteristics of all GUTS participants who completed the baseline questionnaires (N = 27,789), the subset of participants who were invited to the validation study (N = 1,000), and participants with medical records that could be included in the current analyses (N = 318). We combined descriptive information for the 201 who agreed to provide medical record access for the current validation study with the 117 who had provided medical records for a previous study, as there was no statistically significant difference between the two groups (data not shown).

The participants included in the current study (N = 318) were on average 33 years old in 2019, with the majority being white (97%) and over half being female (56%). The participants had an average body mass index (BMI) of 28 kg/m$^2$ in 2019. Almost half of them had been on anti-hypertensive medication (43%). The population characteristics of the individuals being invited to the validation study (N = 1,000) were in general comparable to those who provided medical records (N = 318), except for the lower anti-hypertensive medication use rate (26%). The overall GUTS population (N = 27,789) were slightly younger, with an average age of 31

**Table 1. Population characteristics of the full Growing Up Today Study (GUTS) cohort (N = 27,789), participants invited to provide medical records (N = 1,000), individuals whose medical records were successfully obtained (N = 318), and subgroups.**

| | All (N = 27,789)[a] | Invited (N = 1,000) | Medical records obtained (N = 318)[b] | Hypertensive base on the medical records (N = 132) | Non-hypertensive base on the medical records (N = 186) | Males (N = 139) | Females (N = 179) |
|---|---|---|---|---|---|---|---|
| Age 2019, y | 31.3 ± 3.5 | 33.4 ± 2.4 | 33.3 ± 2.5 | 32.8 ± 3.0 | 33.6 ± 2.0 | 33.3 ± 2.5 | 33.2 ± 2.5 |
| BMI 2019[c], kg/m² | 26.2 ± 5.9 | 28.6 ± 7.1 | 28.3 ± 7.8 | 31.1 ± 9.4 | 26.4 ± 6.0 | 27.3 ± 5.7 | 29.0 ± 9.1 |
| Sex, Male (%) | 12,756 (45.9) | 482 (48.2) | 139 (43.7) | 59 (44.7) | 80 (43.0) | 139 (100) | 0 (0) |
| Race, White (%) | 26,073 (94.7) | 936 (94.0) | 308 (96.9) | 128 (97.0) | 180 (96.8) | 135 (97.1) | 173 (96.6) |
| Overweight in 2019 (%)[d] | 3,222 (29.4) | 298 (32.1) | 96 (31.5) | 37 (30.6) | 59 (32.1) | 51 (39.2) | 45 (25.7) |
| Obese in 2019 (%)[d] | 2,149 (19.6) | 303 (32.7) | 89 (29.2) | 53 (43.8) | 36 (19.6) | 31 (23.8) | 58 (33.1) |
| Anti-hypertensive medication usage ever (%)[d] | 1,133 (4.1) | 256 (25.6) | 136 (42.8) | 116 (87.9) | 20 (10.8) | 62 (44.6) | 74 (41.3) |
| Self-reported blood pressure (mm Hg) | | | | | | | |
| Typical recent systolic[e] | 116 ± 12 | 121 ± 14 | 120 ± 13 | 126 ± 13 | 116 ± 11 | 123 ± 10 | 118 ± 14 |
| Typical recent diastolic[f] | 73 ± 10 | 77 ± 11 | 76 ± 11 | 81 ± 13 | 73 ± 9 | 78 ± 8 | 75 ± 14 |
| Pre-diagnosis systolic[g] | 145 ± 21 | 147 ± 20 | 151 ± 24 | 152 ± 24 | 135 ± 17 | 155 ± 29 | 148 ± 17 |
| Pre-diagnosis diastolic[h] | 91 ± 14 | 94 ± 14 | 95 ± 16 | 96 ± 16 | 86 ± 8 | 92 ± 17 | 98 ± 15 |
| Recorded blood pressure (mm Hg) | | | | | | | |
| Typical recent systolic[i] | NA | NA | 122 ± 14 | 136 ± 13 | 119 ± 12 | 125 ± 13 | 118 ± 14 |
| Typical recent diastolic[i] | NA | NA | 76 ± 10 | 83 ± 11 | 74 ± 9 | 77 ± 9 | 75 ± 11 |
| Pre-diagnosis systolic[j] | NA | NA | 148 ± 13 | 148 ± 13 | NA | 148 ± 12 | 148 ± 14 |
| Pre-diagnosis diastolic[j] | NA | NA | 94 ± 10 | 94 ± 10 | NA | 94 ± 9 | 95 ± 11 |

[a] All participants who completed the baseline questionnaires

[b] Consisting of 201 medical records from the 1000 invited participants and 117 records available from a previous study. We combined the two groups as there was no statistically significant difference between the groups

[c] 10,947 (39.4%) available in all participants; 927 (92.7%) available in invited participants; 305 (95.9%) available in participants with medical records obtained; 121 (91.7%) available in hypertensive participants; 184 (98.9%) available in non-hypertensive participants; 130 (93.5%) available in males; 175 (97.8%) available in females

[d] Percentage in GUTS participants who completed the 2019 questionnaire

[e] 6,108 (22.0%) available in all participants; 856 (85.6%) available in invited participants; 289 (90.9%) available in participants with medical records obtained; 110 (83.3%) available in hypertensive participants; 179 (96.2%) available in non-hypertensive participants; 125 (89.9%) available in males; 164 (91.6%) available in females

[f] 6,109 (22.0%) available in all participants; 854 (85.4%) available in invited participants; 289 (90.9%) available in participants with medical records obtained; 110 (83.3%) available in hypertensive participants; 179 (96.2%) available in non-hypertensive participants; 125 (89.9%) available in males; 164 (91.6%) available in females

[g] 227 (0.8%) available in all participants; 120 (12.0%) available in invited participants; 62 (19.5%) available in participants with medical records obtained; 57 (43.2%) available in hypertensive participants; 5 (2.7%) available in non-hypertensive participants; 30 (21.6%) available in males; 32 (17.9%) available in females

[h] 228 (0.8%) available in all participants; 121 (12.1%) available in invited participants; 63 (19.8%) available in participants with medical records obtained; 58 (43.9%) available in hypertensive participants; 5 (2.7%) available in non-hypertensive participants; 30 (21.6%) available in males; 33 (18.4%) available in females

[i] 192 (60.4%) available in participants with medical records obtained; 34 (25.8%) available in hypertensive participants; 158 (84.9%) available in non-hypertensive participants; 88 (63.3%) available in males; 104 (58.1%) available in females

[j] 61 (19.2%) available in participants with medical records obtained; 61 (46.2%) available in hypertensive participants; 29 (20.9%) available in males; 32 (17.9%) available in females

NA = not available

years in 2019, had lower BMI of 26 kg/m$^2$ in 2019, and had much lower proportions of antihypertensive medication use (4%).

Participants with a diagnosis of hypertension based on the medical record had a higher BMI of 31 kg/m$^2$ and higher rate of antihypertensive medication use of 88%, compared to those without this diagnosis. Twenty participants were evaluated as not having hypertension based on medical records but did report antihypertensive medication use; fifteen of these had self-reported hypertension diagnosis and five did not. For most of these patients, a hypertensive blood pressure was recorded only on one visit or only during pregnancy. Population characteristics were similar between male and female participants. Of those with hypertension, 19 (14%) were diagnosed between 2018 and 2020 and 113 (86%) before 2018 when the US guidelines on the definition of hypertension changed.

The agreement between self-reported hypertension and diagnosis based upon medical records was 85.5% (Table 2). The kappa value of 0.72 suggested substantial agreement between hypertension status from self-reporting and medical records. Sensitivity of 100% showed that all hypertensive participants were correctly identified by self-report, while specificity showed that 75.3% of non-hypertensive participants reported their condition correctly. No statistically significant difference was found between male and female participants in terms of the validity of self-reported hypertension. Among the 28 female participants who self-reported as hypertensive but this was not confirmed by medical record review, seven were diagnosed with hypertension only transiently during pregnancy. After excluding these seven female participants from the analysis, specificity in females increased from 73.6% to 78.8%, and the overall agreement increased from 85.5% to 87.5%.

Office measurements of blood pressure were available for 186 participants (58.5%) who had self-reported a recent blood pressure and for 28 participants (21.2%) who had self-reported their blood pressure prior to diagnosis (Table 3). The self-reported typical recent blood pressures were on average 118/74 mm Hg, lower than those extracted from medical records, averaging 121/76 mm Hg. The correlations between self-reported blood pressures and those from medical records were moderate for typical recent blood pressures, with Spearman's r coefficients of 0.54 and 0.43 for systolic and diastolic blood pressures, respectively. The correlations were low for pre-diagnosis blood pressures, with Spearman's r coefficients below 0.3. The absolute differences in blood pressure readings between self-report and medical records were small, with average differences of 3.5/1.2 mm Hg in typical recent blood pressures. Most of the CIs included 0, indicating the differences were not statistically significant. Similar patterns were observed in the male and female participants, respectively.

**Table 2. Validity of self-reported hypertension among 318 participants in the GUTS cohort.**

|  | All | Males | Females | P-value* |
|---|---|---|---|---|
| True-positive, N | 132 | 59 | 73 | - |
| True-negative, N | 140 | 62 | 78 | - |
| False-positive, N | 46 | 18 | 28 | - |
| False-negative, N | 0 | 0 | 0 | - |
| Agreement (%) | 85.5 | 87.1 | 84.4 | 0.61 |
| Kappa | 0.72 | 0.75 | 0.69 | - |
| Sensitivity (%) | 100 | 100 | 100 | - |
| Specificity (%) | 75.3 | 77.5 | 73.6 | 0.66 |

*Difference between males and females tested by Chi-squared test or Fisher's exact test as appropriate. Two-sided P-values with P<0.05 were used to determine statistical significance.

**Table 3. Validity of self-reported blood pressure measurements (mm Hg) among study participants in the GUTS study.**

| | Self-reported | Medical record | Pearson's r | Spearman's r | Difference (95% CI) |
|---|---|---|---|---|---|
| **All** | | | | | |
| Typical recent systolic (N = 186) | 118 ± 12 | 121 ± 14 | 0.43 | 0.54 | -3.5 (-5.5, -1.5) |
| Typical recent diastolic (N = 186) | 74 ± 11 | 76 ± 10 | 0.33 | 0.43 | -1.2 (-3.0, 0.5) |
| Pre-diagnosis systolic (N = 28) | 155 ± 29 | 148 ± 13 | 0.32 | 0.23 | 6.6 (-4.0, 17.2) |
| Pre-diagnosis diastolic (N = 29) | 95 ± 16 | 96 ± 12 | 0.26 | 0.18 | -0.9 (-7.6, 5.7) |
| **Males** | | | | | |
| Typical recent systolic (N = 83) | 121 ± 9 | 125 ± 13 | 0.35 | 0.46 | -4.2 (-7.0, -1.4) |
| Typical recent diastolic (N = 83) | 77 ± 8 | 77 ± 9 | 0.29 | 0.28 | -0.1 (-2.3, 2.1) |
| Pre-diagnosis systolic (N = 14) | 157 ± 38 | 148 ± 11 | 0.34 | 0.03 | 8.4 (-12.4, 29.3) |
| Pre-diagnosis diastolic (N = 14) | 92 ± 22 | 95 ± 12 | 0.42 | 0.19 | -3.3 (-14.7, 8.2) |
| **Females** | | | | | |
| Typical recent systolic (N = 103) | 115 ± 14 | 118 ± 14 | 0.41 | 0.56 | -2.9 (-5.8, 0.0) |
| Typical recent diastolic (N = 103) | 72 ± 13 | 75 ± 11 | 0.34 | 0.49 | -2.2 (-4.8, 0.5) |
| Pre-diagnosis systolic (N = 14) | 153 ± 14 | 148 ± 16 | 0.48 | 0.39 | 4.7 (-4.3, 13.7) |
| Pre-diagnosis diastolic (N = 15) | 98 ± 9 | 96 ± 12 | -0.05 | 0.05 | 1.3 (-7.3, 9.8) |
| **Hypertensive** | | | | | |
| Typical recent systolic (N = 32) | 127 ± 14 | 136 ± 12 | 0.06 | 0.01 | -8.2 (-14.6, -1.8) |
| Typical recent diastolic (N = 32) | 84 ± 14 | 83 ± 11 | -0.11 | 0.04 | 1.2 (-5.7, 8.1) |
| Pre-diagnosis systolic (N = 28) | 155 ± 29 | 148 ± 13 | 0.32 | 0.23 | 6.6 (-4.0, 17.2) |
| Pre-diagnosis diastolic (N = 29) | 95 ± 16 | 96 ± 12 | 0.26 | 0.18 | -0.9 (-7.6, 5.7) |
| **Non-hypertensive** | | | | | |
| Typical recent systolic (N = 154) | 116 ± 11 | 118 ± 12 | 0.37 | 0.49 | -2.5 (-4.6, -0.4) |
| Typical recent diastolic (N = 154) | 72 ± 9 | 74 ± 9 | 0.36 | 0.39 | -1.7 (-3.4, -0.1) |

Among hypertensive participants, self-reported typical recent systolic blood pressures were on average 8.2 mm Hg lower than those from medical records, whereas the pre-diagnosis systolic blood pressures were on average 6.6 mm Hg higher in self-reports. Among non-hypertensive participants, the average difference in typical recent systolic blood pressures was 2.5 mm Hg. The differences in diastolic blood pressures were similar for hypertensive and non-hypertensive participants. The Bland-Altman plot confirms a good agreement between blood pressure measurements obtained from medical records and those self-reported through questionnaires (Fig 2). This is indicated by the estimated biases being close to zero and the presence of only a few outliers.

## Discussion

In this representative sample of GUTS young adults, we found a high accuracy of self-reported hypertension validated by medical records – all participants who were diagnosed with hypertension by their doctors self-reported this diagnosis correctly (i.e., 100% sensitivity) and 75% of the participants who were not diagnosed with hypertension correctly reported that they did not have hypertension (i.e., 75% specificity). Although the absolute differences in blood pressure measurements between self-report and medical records were small, these measures were only moderately correlated.

The sensitivity of self-reported hypertension in our study is higher than most previous studies. A 2018 systematic review reported that the sensitivity of self-reported hypertension ranged from 13% to 92% across 22 studies [4]. The highest sensitivity, 92%, was reported in a subset of 1,114 participants from the US National Health and Examination Survey (NHANES) aged 25

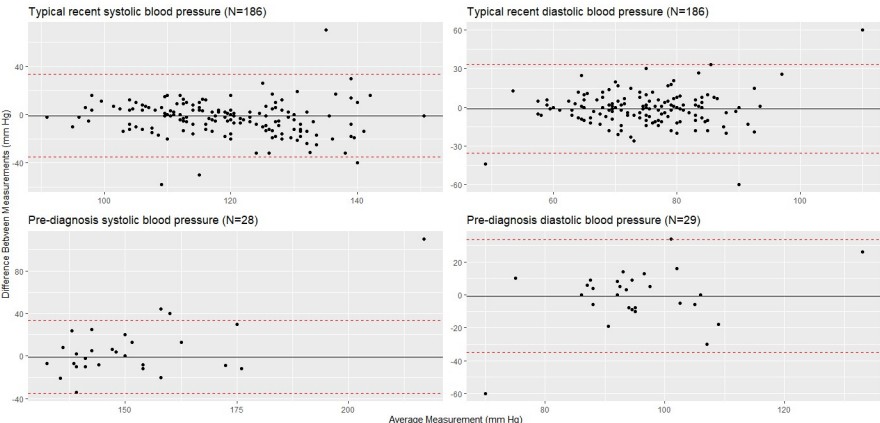

**Fig 2. Bland-Altman Plot of agreement between blood pressure measurements.**

years and older and living in urban metropolitan areas [15]. This study further reported a sensitivity close to 100% in a sub-population aged 35 years and older. The sensitivity of self-reported hypertension was much lower in the general NHANES population, without restricting it to the urban settings, at 71% [16]. Living in urban areas has been associated with higher accuracy of self-reporting because of higher educational rate compared to living in rural areas [17]. The higher sensitivity observed in the current study may reflect better medical awareness of the study population compared with the general population, as all participants were offspring of nurses. Indeed, sensitivity was 94% in a comparison of self-report to medical record review in NHSII, all registered nurses and mothers of the GUTS participants aged 32 to 52 years [18].

The specificity of 75% in the current study is among the lowest in the reported range of 72% to 98.8% in the 2018 systematic review [4]. Several factors likely contributed to a lower specificity among the GUTS participants. We used a relatively strict definition of hypertension diagnosis when reviewing the medical records. While often diagnosed in outpatient clinics on the basis of one office blood pressure, hypertension cannot be correctly diagnosed without repeated hypertensive readings on separate visits [11]. Therefore, we did not consider participants as hypertensive if a physician indicated hypertension based on high blood pressure only on one visit. White coat hypertension or the white coat effect is quite common [19], yet participants may have recalled being told their blood pressure was high at an office visit and interpreted that to mean a diagnosis of hypertension. Thus, several factors, e.g., stress and concurrent illness, may have contributed to lower specificity. Furthermore, since our focus was on defining chronic hypertension, we considered participants who only had transient pregnancy-related hypertension as non-hypertensive. Yet the question asked on the GUTS questionnaire did not distinguish pregnancy-related hypertension from chronic hypertension. This may also explain the slightly better validity in males than in females observed in the current study, which was different from most previous studies that reported higher validity in females [16, 20, 21]. The validity in females was similar to that in males after excluding the female participants who reported hypertension but were only diagnosed during pregnancy.

Our results also reflect the difficulty in determining hypertension diagnosis by reviewing medical records. Diagnosing hypertension is nuanced and not always straightforward, because hypertensive blood pressures are not always persistent [22]. Further, confirming the diagnosis of hypertension in this study was not possible when blood pressure readings were unavailable

or information was available from one visit only. This may have also contributed to the misclassification of hypertension cases when reviewing medical records. Moreover, physicians often used terminology such as 'elevated blood pressure' or 'prehypertension' that may be interpreted as hypertension by the patient.

We observed better agreement of office measured and self-reported blood pressure in non-hypertensive participants than in hypertensive participants. This may be explained by the relatively stable blood pressure in individuals without hypertension, compared to more varied blood pressure in individuals with hypertension, depending on how well the high blood pressure is controlled. Our results also identified the challenge of low response rates and incomplete reports of blood pressure measurements.

## Strengths and limitations

One of the strengths of this study is the thorough review of medical records and diagnoses by an experienced clinical hypertension specialist. We were additionally able to extract blood pressure measurements from medical records to compare with self-reported readings. However, the blood pressure records were incomplete. The sample size included in the current analyses is modest, and thus hampered more detailed exploration of demographic factors. The percentage of participants agreeing to have their medical records released is also relatively small. However, there did not appear to be evidence of selection bias between the participants who provided medical records and the full cohort, suggesting that these findings may be generalizable to the entire cohort. We acknowledge that reviewing medical records may not be the best way to verify participants' hypertension status. Participants were only able to have a hypertension diagnosis on medical records if they had regular healthcare visits. Therefore, identifying hypertensive participants with self-reported diagnosis is likely to underestimate the incidence in the current study population and the entire cohort. However, this would not affect the validity of self-reported hypertension diagnosis. The overrepresentation of white individuals who are offspring of nurses in the study cohort may hamper the generalizability of the findings to the broader population.

## Conclusions

Validity of self-reported hypertension was high in GUTS, ensuring use as an endpoint in future studies with confidence. Blood pressure readings, although only showing small absolute differences, were incompletely reported and only correlated moderately with medical records, and therefore should be used with caution. Importantly, we have demonstrated that young adults likely without formal medical training are able to report hypertension status with reasonable accuracy.

## Acknowledgments

We thank the participants in the Growing Up Today Study as well as their mothers participating in the Nurses' Health Study II.

## Author Contributions

**Conceptualization:** Jaime E. Hart, Francine Laden.

**Formal analysis:** Jie Chen.

**Funding acquisition:** Francine Laden.

**Methodology:** Jie Chen.

**Project administration:** Jaime E. Hart.

**Supervision:** Jaime E. Hart, Naomi D. L. Fisher, Francine Laden.

**Validation:** Jaime E. Hart.

**Writing – original draft:** Jie Chen.

**Writing – review & editing:** Jaime E. Hart, Naomi D. L. Fisher, Francine Laden.

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
