## [Decision Letter · Decision Letter 0]

30 May 2024

PONE-D-23-34197Validation of self-reported hypertension in young adults in the US-based Growing Up Today Study (GUTS)PLOS ONE

Dear Dr. Chen,

Thank you for submitting your manuscript to PLOS ONE. After careful consideration, we feel that it has merit but does not fully meet PLOS ONE’s publication criteria as it currently stands. Therefore, we invite you to submit a revised version of the manuscript that addresses the points raised during the review process.

We look forward to receiving your revised manuscript.

Kind regards,

Ibrahim Sebutu Bello, MBBS, MPH, MD, FMCGP

Academic Editor

PLOS ONE

Additional Editor Comments:

Introduction: The introduction is scanty. To provide more context and gaps for the current study, authors may review previous studies done within and outside of the US. There are many published works in this area of research.

Page 6, Lines 101-102

Authors need to clarify this statement: “We compared the average difference between blood pressure measurements self-reported on the 2019 questionnaire and those recorded in the medical records, using paired T-tests.” (see also lines 11-13 in the Abstract)

It appears the comparison was done using correlations.

References: Please ensure all the references align with the journal format, for example, references 4 and 7.

Reviewers' comments:

Reviewer's Responses to Questions

**Comments to the Author**

1. Is the manuscript technically sound, and do the data support the conclusions?

Reviewer #1: Yes

2. Has the statistical analysis been performed appropriately and rigorously? 

Reviewer #1: Yes

3. Have the authors made all data underlying the findings in their manuscript fully available?

Reviewer #1: Yes

4. Is the manuscript presented in an intelligible fashion and written in standard English?

Reviewer #1: Yes

5. Review Comments to the Author

Reviewer #1: Thank you for conducting this study. It is a truly necessary one. However, here are some comments that could help improve the manuscript;

Introduction:

1. It would be beneficial if you add more body to the overview of your topic "self reported hypertension" in the introduction.

2. Lines 24 & 25; State some statistics clearing regarding the validity of self reported hypertension as several studies have examined this before just in different populations. But it would give a better background here.

3. Lines 30-34; Summarize the objective section into one clear statement of your study objective.

Methods

Your methods are comprehensively described. However,

4. Lines 37-86; As you are trying to describe the basis of your study, the methods used in the GUTS are so mixed up with yours. You should describe your methods clearly and summarise how GUTS was done in maybe 2-4 lines with clear citation as you have done.

Results

5. Kindly revise this results section and summarise the key features of your participants and the key findings as per ypur objectives clearly stating the figures, rather than describing evrything in the table (which is a very clear representaion of your data).

Discussion

6. Lines 183-191; This paragraph should in a succinct manner re-echo your objective(s), and state the very key findings in your study to get the reader to be on the same page as you are.

7. Lines 193-200; There are more studies that examined the validity of hypertension whose you can compare with yours in addtion to the ones stated in your manuscript.

6. PLOS authors have the option to publish the peer review history of their article (what does this mean?). If published, this will include your full peer review and any attached files.

Reviewer #1: **Yes: **Dr. Winnie Kibone

---

## [Author Response · Author response to Decision Letter 0]

24 Jun 2024

We are grateful to the Academic Editor and the Reviewer for their thoughtful and thorough review of our manuscript and for their supportive response to our work. We offer the following point-by-point responses to the individual comments. We note that throughout this response document, quoted line numbers pertain to the marked copy of the revised manuscript. We hope that our responses will satisfy the Editor and Reviewer’s concerns.

Editor Comments:

Introduction: The introduction is scanty. To provide more context and gaps for the current study, authors may review previous studies done within and outside of the US. There are many published works in this area of research.

We have now expanded the Introduction section to provide more context and research gaps. The revised text reads:

“Self-reporting is a simple and low-cost method that is often used in large epidemiologic studies for identifying participants with hypertension [1, 2]. The validity of self-reported hypertension can be influenced by demographic, socioeconomic, and cultural factors [3-6]. A meta-analysis of 11 studies found that self-report correctly identified individuals with hypertension in 42% (95% confidence interval [CI], 31−54) of hypertensive cases and correctly identified individuals without hypertension in 90% (95% CI 84−93) of non-cases [4]. Large heterogeneity was found across individual studies. Using the India National Family Health Survey data, researchers found self-reported hypertension has higher sensitivity among individuals aged 35–49 years than those aged 15–34 years, while specificity was higher among the younger individuals [6]. The younger adults may differ from the older populations in many aspects that could affect the validity of self-reporting, such as frequency of doctor visits and awareness of health status. While the majority of the previous research has been conducted in older populations [5, 7], the validity of self-reported hypertension in young adults has not been sufficiently assessed. 

The Growing Up Today Study (GUTS) was established to study the health of adolescents and young adults [8]. GUTS I and GUTS II (known collectively as GUTS) were founded in 1996 and 2004, respectively, by inviting mothers in the ongoing Nurses’ Health Study II (NHS II) to enroll their children who were then between the ages of 9 and 14 (GUTS I) or 10 and 17 (GUTS II) [9]. More than 27,000 participants across the US completed and returned the baseline questionnaires. Follow-up questionnaires were sent to participants annually. The cohorts were combined in 2013 when all participants had reached age 18. Approximately 86% of the females and 83% of the males have returned at least one follow-up questionnaire. Self-reported hypertension has been asked on all questionnaires since 2010. To use self-reported hypertension data with confidence, it is necessary to examine its validity. 

Therefore, we aimed to validate hypertension diagnoses and blood pressure measurements from self-reported questionnaires using medical records among a sample of participants in GUTS. In the current study, we included follow-up data up to the end of 2019, covering the period of transition in the US national definitions of hypertension from 140/90 millimeters of mercury (mm Hg) to 130/80 mm Hg [10, 11].” (lines 23−59)

Page 6, Lines 101-102

Authors need to clarify this statement: “We compared the average difference between blood pressure measurements self-reported on the 2019 questionnaire and those recorded in the medical records, using paired T-tests.” (see also lines 11-13 in the Abstract)

It appears the comparison was done using correlations.

In addition to assessing the correlations between measurements from two sources, we compared the measurements using paired t-tests. We reported the mean difference and 95% CIs in Table 3. A 95% CI not including zero indicates a statistically significant difference between blood pressure readings. We have now modified the text for clarity.

“We assessed the correlations between blood pressure measurements self-reported on the 2019 questionnaire and those from medical records and compared the measurements using paired t-tests.” (lines 11−13 in the Abstract)

“We assessed the correlations between blood pressure readings self-reported on the 2019 questionnaire and those recorded in the medical records using both Pearson and Spearman’s correlation coefficients. We compared blood pressure readings from two sources using paired t-tests. We reported the mean difference and 95% confidence intervals (CIs) based on an α=0.05. A 95% CI not including zero indicates a statistically significant difference between blood pressure readings.” (lines 141−147 in the Methods)

References: Please ensure all the references align with the journal format, for example, references 4 and 7.

We have checked throughout the manuscript to ensure the references and other formatting meet the journal requirements.

Reviewers' comments:

Reviewer #1: Thank you for conducting this study. It is a truly necessary one. However, here are some comments that could help improve the manuscript;

Introduction:

1. It would be beneficial if you add more body to the overview of your topic "self reported hypertension" in the introduction.

We have now expanded the Introduction section to provide more context and research gaps. The revised text reads:

“Self-reporting is a simple and low-cost method that is often used in large epidemiologic studies for identifying participants with hypertension [1, 2]. The validity of self-reported hypertension can be influenced by demographic, socioeconomic, and cultural factors [3-6]. A meta-analysis of 11 studies found that self-report correctly identified individuals with hypertension in 42% (95% confidence interval [CI], 31−54) of hypertensive cases and correctly identified individuals without hypertension in 90% (95% CI 84−93) of non-cases [4]. Large heterogeneity was found across individual studies. Using the India National Family Health Survey data, researchers found self-reported hypertension has higher sensitivity among individuals aged 35–49 years than those aged 15–34 years, while specificity was higher among the younger individuals [6]. The younger adults may differ from the older populations in many aspects that could affect the validity of self-reporting, such as frequency of doctor visits and awareness of health status. While the majority of the previous research has been conducted in older populations [5, 7], the validity of self-reported hypertension in young adults has not been sufficiently assessed. 

The Growing Up Today Study (GUTS) was established to study the health of adolescents and young adults [8]. GUTS I and GUTS II (known collectively as GUTS) were founded in 1996 and 2004, respectively, by inviting mothers in the ongoing Nurses’ Health Study II (NHS II) to enroll their children who were then between the ages of 9 and 14 (GUTS I) or 10 and 17 (GUTS II) [9]. More than 27,000 participants across the US completed and returned the baseline questionnaires. Follow-up questionnaires were sent to participants annually. The cohorts were combined in 2013 when all participants had reached age 18. Approximately 86% of the females and 83% of the males have returned at least one follow-up questionnaire. Self-reported hypertension has been asked on all questionnaires since 2010. To use self-reported hypertension data with confidence, it is necessary to examine its validity. 

Therefore, we aimed to validate hypertension diagnoses and blood pressure measurements from self-reported questionnaires using medical records among a sample of participants in GUTS. In the current study, we included follow-up data up to the end of 2019, covering the period of transition in the US national definitions of hypertension from 140/90 millimeters of mercury (mm Hg) to 130/80 mm Hg [10, 11].” (lines 23−59)

2. Lines 24 & 25; State some statistics clearing regarding the validity of self reported hypertension as several studies have examined this before just in different populations. But it would give a better background here.

We have included more numeric descriptions in the background.

“A meta-analysis of 11 studies found that self-report correctly identified individuals with hypertension in 42% (95% confidence interval [CI], 31−54) of hypertensive cases and correctly identified individuals without hypertension in 90% (95% CI 84−93) of non-cases [4]. Large heterogeneity was found across individual studies.” (lines 26−30)

3. Lines 30-34; Summarize the objective section into one clear statement of your study objective.

We have rewritten the objective section.

“Therefore, we aimed to validate hypertension diagnoses and blood pressure measurements from self-reported questionnaires using medical records among a sample of participants in GUTS. In the current study, we included follow-up data up to the end of 2019, covering the period of transition in the US national definitions of hypertension from 140/90 millimeters of mercury (mm Hg) to 130/80 mm Hg [10, 11].” (lines 55−59)

Methods

Your methods are comprehensively described. However,

4. Lines 37-86; As you are trying to describe the basis of your study, the methods used in the GUTS are so mixed up with yours. You should describe your methods clearly and summarise how GUTS was done in maybe 2-4 lines with clear citation as you have done.

To avoid confusion and provide more context for the current study, we have now shortened and moved the overall study design description of GUTS to the Introduction section. We also rearranged the study design section. 

“Study design

We identified a random sample of 500 participants who ever reported being diagnosed with hypertension, and a random sample of 500 participants who never reported a hypertension diagnosis, based on questionnaires from 2010 to 2019. We asked the 1000 participants for permission to access their medical records to confirm the self-reported information. Of the 1000 participants invited, 201 (20.1%) provided written consent to the study and their medical records were scanned for review. All participants were recruited prospectively from June 23rd 2020 to March 9th 2022. An additional 117 GUTS participants previously provided written consent to provide their medical records for hypertension status confirmation, and thus paired self-reported and medical record information was available for 318 participants in the current analyses. This study was approved by the Institutional Review Board at the Brigham and Women’s Hospital and participants provided informed written consent.

Hypertension was self-reported on all questionnaires from 2010 to 2019. The question asks, “Have you ever been diagnosed with high blood pressure or hypertension?" which was adapted from a question asked in Wave III of the National Longitudinal Study of Adolescent to Adult Health (Add Health) [12]. Using this question, hypertension prevalence was estimated to be 5.6% in 15,187 young adult (18-26 years) participants in Add Health, which was close to estimates from studies where classification of hypertension was based on measured blood pressure [13]. As a follow-up question, GUTS participants were asked to provide the year of first diagnosis. In the 2019 questionnaire, participants were further asked to recall their most recently measured (within 2 years) systolic and diastolic blood pressure. Participants who were on blood pressure lowering medications were additionally asked to recall their office blood pressure prior to starting medication.” (lines 87−108)

Results

5. Kindly revise this results section and summarise the key features of your participants and the key findings as per ypur objectives clearly stating the figures, rather than describing evrything in the table (which is a very clear representaion of your data).

We have reorganized our Results section to align with the objectives and have included more numerical data in the description of the results.

“The participants included in the current study (N=318) were on average 33 years old in 2019, with the majority being white (97%) and over half being female (56%). The participants had an average body mass index (BMI) of 28 kg/m2 in 2019. Almost half of them had been on anti-hypertensive medication (43%). The population characteristics of the individuals being invited to the validation study (N=1,000) were in general comparable to those who provided medical records (N=318), except for the lower anti-hypertensive medication use rate (26%). The overall GUTS population (N=27,789) were slightly younger, with an average age of 31 years in 2019, had lower BMI of 26 kg/m2 in 2019, and had much lower proportions of antihypertensive medication use (4%).

Participants with a diagnosis of hypertension based on the medical record had a higher BMI of 31 kg/m2 and higher rate of antihypertensive medication use of 88%, compared to those without this diagnosis. Twenty participants were evaluated as not having hypertension based on medical records but did report antihypertensive medication use; fifteen of these had self-reported hypertension diagnosis and five did not. For most of these patients, a hypertensive blood pressure was recorded only on one visit or only during pregnancy. Population characteristics were similar between male and female participants. Of those with hypertension, 19 (14%) were diagnosed between 2018 and 2020 and 113 (86%) before 2018 when the US guidelines on the definition of hypertension changed.” (lines 194−209)

“Office measurements of blood pressure were available for 186 participants (58.5%) who had self-reported a recent blood pressure and for 28 participants (21.2%) who had self-reported their blood pressure prior to diagnosis (Table 3). The self-reported typical recent blood pressures were on average 118/74 mm Hg, lower than those extracted from medical records, averaging 121/76 mm Hg. The correlations between self-reported blood pressures and those from medical records were moderate for typical recent blood pressures, with Spearman’s r coefficients of 0.54 and 0.43 for systolic and diastolic blood pressures, respectively. The correlations were low for pre-diagnosis blood pressures, with Spearman’s r coefficients below 0.3. The absolute differences in blood pressure readings between self-report and medical records were small, with average differences of 3.5/1.2 mm Hg in typical recent blood pressures. Most of the CIs included 0, indicating the differences were not statistically significant. Similar patterns were observed in the male and female participants, respectively.” (lines 224−236)

“Among hypertensive participants, self-reported typical recent systolic blood pressures were on average 8.2 mm Hg lower than those from medical records, whereas the pre-diagnosis systolic blood pressures were on average 6.6 mm Hg higher in self-reports. Among non-hypertensive participants, the average difference in typical recent systolic blood pressures was 2.5 mm Hg. The differences in diastolic blood pressures were similar for hypertensive and non-hypertensive participants. The Bland-Altman plot confirms a good agreement between blood pressure measurements obtained from medical records and those self-reported through questionnaires (Fig 2). This is indicated by the estimated biases being close to zero and the presence of only a few outliers.” (lines 250−257)

Discussion

6. Lines 183-191; This paragraph should in a succinct manner re-echo your objective(s), and state the very key findings in your study to get the reader to be on the same page as you are.

We have rewritten the paragraph.

“In this representative sample of GUTS young adults, we found a high accuracy of self-reported hypertension validated by medical records − all participants who were diagnosed with hypertension by their doctors self-reported this diagnosis correctly (i.e., 100% sensitivity) and 75% of the participants who were not diagnosed with hypertension correctly reported that they did not have hypertension (i.e., 75% specificity). Although the absolute differences in blood pressure measurements between self-report and medical records were small, these measures were only moderately correlated.” (lines 262−271)

7. Lines 193-200; There are more studies that examined the vali

---

## [Decision Letter · Decision Letter 1]

25 Sep 2024

Validation of self-reported hypertension in young adults in the US-based Growing Up Today Study (GUTS)

PONE-D-23-34197R1

Dear Dr. Chen,

We’re pleased to inform you that your manuscript has been judged scientifically suitable for publication and will be formally accepted for publication once it meets all outstanding technical requirements.

Kind regards,

Ibrahim Sebutu Bello, MBBS, MPH, MD, FMCGP

Academic Editor

PLOS ONE

Additional Editor Comments (optional):

NIL

Reviewers' comments:

Reviewer's Responses to Questions

**Comments to the Author**

1. If the authors have adequately addressed your comments raised in a previous round of review and you feel that this manuscript is now acceptable for publication, you may indicate that here to bypass the “Comments to the Author” section, enter your conflict of interest statement in the “Confidential to Editor” section, and submit your "Accept" recommendation.

Reviewer #2: All comments have been addressed

2. Is the manuscript technically sound, and do the data support the conclusions?

Reviewer #2: Yes

3. Has the statistical analysis been performed appropriately and rigorously? 

Reviewer #2: Yes

4. Have the authors made all data underlying the findings in their manuscript fully available?

Reviewer #2: Yes

5. Is the manuscript presented in an intelligible fashion and written in standard English?

Reviewer #2: No

6. Review Comments to the Author

Reviewer #2: the article will be benefited from a proof reading service. I have no issue with other matters like about dual publication, research ethics, or publication ethics.

7. PLOS authors have the option to publish the peer review history of their article (what does this mean?). If published, this will include your full peer review and any attached files.

Reviewer #2: No

---

## [Editor Report · Acceptance letter]

15 Oct 2024

PONE-D-23-34197R1 

PLOS ONE

Dear Dr. Chen, 

I'm pleased to inform you that your manuscript has been deemed suitable for publication in PLOS ONE. Congratulations! Your manuscript is now being handed over to our production team.

Kind regards, 

on behalf of

Dr. Ibrahim Sebutu Bello 

Academic Editor

PLOS ONE